# Predictors of recovery rate among undernourished HIV-positive adults treated with ready-to-use therapeutic food at Debre Markos Comprehensive Specialized Hospital: A retrospective cohort study

Habtamu Gebremeskel Woldie[1], Daniel Bekele Ketema[2]*, Mulatu Ayana[3], Animut Alebel[4,5]

1 Department of Hospital Pharmacy, Debre Markos Comprehensive Specialized Hospital, Debre Markos, Ethiopia, 2 Department of Public Health, College of Health Science, Debre Markos University, Debre Markos, Ethiopia, 3 College of Medicine and Health Science, Ambo University, Ambo, Ethiopia, 4 Department of Nursing, College of Health Science, Debre Markos University, Debre Markos, Ethiopia, 5 Faculty of Health, School of Public Health, University of Technology Sydney, Sydney, Australia

* dabekle121@gmail.com

## Abstract

### Background

Nutritional support is becoming more widely acknowledged as a crucial component of the key package of care for HIV/AIDS patients. This nutritional support is exceptionally important for patients in sub-Saharan Africa, including Ethiopia, where HIV/AIDS is very prevalent. However, there is a lack of evidence on the recovery rate and predictors at the study site and at large in Ethiopia. Therefore, this study will give some insight into the recovery rate and its predictors among under-nourished HIV-positive adults treated with Ready to Use Therapeutic Food (RUTF) attending at Debre Markos referral hospital. Moreover, the findings of this study will be used by both governmental and non-governmental organizations to allocate more resources to mitigate the nutritional problems for people living with HIV.

### Methods

An institution-based retrospective cohort study was conducted among 453 under-nourished HIV positive adults treated with RUTF at Debre Markos referral Hospital from the 1st of July, 2015 to the 31st of December, 2017. The study participants were selected using a simple random sampling technique. Data were extracted from patient charts using a standardized data extraction checklist. Data were entered into Epi-Data Version 4.2 and analyzed using Stata Version 14. The Kaplan-Meier survival curve was used to estimate the time to recovery. Log-rank test was used to compare the recovery time between different baseline categorical variables. The bivariable and multivariable Cox-proportional hazard regression models were fitted for potential predictors of recovery time. Adjusted hazard Ratios (AHRs)

**Data Availability Statement:** All relevant data are within the manuscript and its Supporting Information files.

**Funding:** The author(s) received no specific funding for this work.

**Competing interests:** The authors have declared that no competing interests exist.

**Abbreviations:** HIV, Human Immunodeficiency Virus; AIDS, Acquired Immune Deficiency Syndrome; SSA, Sub-Saharan Africa; RUTF, Reedy to Use Therapeutic Food; MAM, Moderate Acute Malnourished; SAM, Severely Acute Malnourished; FNTA, Food and Nutrition Technical Assistant; CPT, Cotrimoxazole Preventive Therapy; ART, Anti-Retroviral therapy; OIs, Opportunistic Infections; WHO, World Health Organization.

with 95% CIs were used to measure the strength of association and test statistical significance.

## Results

A total of 453 undernourished HIV-positive adults were included in the final analysis. About 201, 44.4% (95%CI: 38.9, 49.0%) patients participating in the RUTF program were recovered based on predetermined exit criteria with incidence of 10.65 (95% CI: 9.28, 12.23) per 100 person-month observations. Being moderately undernourished (AHR: 11.0, 95% CI: 5.3, 23.1), WHO clinical stage (I or II) (AHR:1.8, 95% CI: 1.2, 2.6), and working functional status at baseline (AHR = 2.34, 95%CI: 1.01,5.45) were predictors of recovery time.

## Conclusion

This study concluded that the overall nutritional recovery rate was below the acceptable minimum requirement which at least 75% of patients should recovered. Mild to moderate undernutrition at baseline, WHO clinical stage I or II at enrolment, and working functional status were found to be predictors of recovery time in HIV/AIDS patients treated with the RUTF. As a result, special attention should be paid to severely malnourished patients, WHO clinical stages III or higher, and patients who are bedridden or ambulatory during treatment.

## Introduction

In 2019, around 38 million people worldwide were infected with HIV with 1.7 million of them were newly infected [1]. According to HIV-related estimations and projections for 2019, about 860, 000 persons in Ethiopia were living with HIV, with adults accounting for the largest share with a prevalence rate of 0.9 percent [2].

HIV and malnutrition are closely linked and have a synergetic effect on one another [3–5]. Malnutrition is dramatically increased by HIV. Malnutrition on the other hand, has been shown to hasten the progression of HIV to AIDS [5,6]. They have the potential to harm the immune system and increase infection susceptibility, morbidity and mortality [3,5]. Undernutrition among HIV-positive people is a major public health concern worldwide, particularly in Sub-Saharan Africa (SSA) [7].

Dietary methods such as RUTF have been demonstrated to be effective in addressing undernutrition in HIV patients [8–11]. RUTF is a Plumpynut®-based paste (Plumpy nut or Plumpy sup) in a plastic wrapper for treatment of under nutrition. Peanut-based paste with sugar, vegetable, fat and skimmed milk are among the constitutes of Plumpy nut or Plumpy sup [12]. This therapeutic food is prescribed for a defined period of time, usually 3–6 months, based on anthropometric admission and exit criteria [13].

Adults with mild to moderate acute malnutrition (MAM) and severely acute malnutrition (SAM) should receive therapeutic food, such as RUTF, which is nutritionally similar to F-100, according to WHO guidelines [14]. The food by prescription program (FBP) provides undernourished HIV positive people with food and nutritional support in the form of therapeutic and supplementary feeding, as well as nutritional assessment and counseling at health facilities. The initiative was created to fill dietary intervention gaps in HIV palliative care and assistance for disadvantaged groups, as well as to supplement the antiretroviral therapy (ART) program [9,15].

FBP has been implemented by the Ethiopian government from 2010 with UNICEF and USAID/Ethiopia collaborating with the Ethiopian Ministry of Health and HIV/AIDS Prevention and Control Office, as well as the Food and Nutrition Technical Assistant (FANTA) program [15]. Adult HIV positive patients with MAM receive two sachets of RUTF daily for a minimum of three and a maximum of four months, while those with SAM receive four sachets daily for a minimum of six months and a maximum of eight months under this component of the program [16].

The recovery rate of patients on RUTF were influenced by a variety of factors. Several studies have found that socio-demographic predictors (gender, age, educational level) [16,17], clinical and immunological factors (WHO stages (stage I and II), absence of opportunistic infection, CD4 cell percentage above the threshold, ART status, and baseline nutritional status) have an effect on the Recovery rate [16,18–21]. However, almost all of these studies did not considered censored time to estimate unbiased population parameter at each instantaneous time. The research presented in this report seeks to contribute to filling these gaps identified in the literatures. Several studies have also provided evidence on the recovery time of HIV/AIDS patients treated with RTUF varied. The median recovery duration was 3.7 months (IQR 2.2–6.1) in a long-term study done at SSA [19]. According to another Kenyan study, the median recovery time was 100 days [20]. In Ethiopia, however, a joint study by Tufts University and Save the Children found that the average recovery period for SAM and MAM patients to restore excellent nutritional status and meet discharge requirements was four months and two months, respectively [16].

Many research on the undernourished HIV positive adults have been undertaken in many regions of the world, and they have found that the problem is particularly prevalent in SSA, including Ethiopia. However, there was a paucity of evidence on the optimal recovery duration and determinants among undernourished HIV positive people in Ethiopia who were treated with RUTF, particularly at the research site. As a result, the recovery rate and its determinants among undernourished HIV positive people treated with RUTF at Debre Markos referral hospital will be explored in this study. Furthermore, the findings of this study will be used by both governmental and non-governmental sectors to devote additional resources to help persons living with HIV with their dietary needs.

## Methods

### Study area and period

An Institution-based retrospective cohort study was conducted at Debre Markos Referral Hospital from the 1st of July, 2015 to the 31st of December, 2017. Debre Markos is the zonal capital of East Gojjam zone. It is located Northwestern Ethiopia, in Amhara Regional State, East Gojjam zone, at a distance of 300 kilometers from Addis Ababa, the capital city of Ethiopia, and 265 kilometers from Bahirdar the regional capital. Debre Markos Referral Hospital provides service for More than five million people in its catchment area [22]. Since the beginning of antiretroviral therapy (ART) at Debre Markos referral hospital, a total of 9,612 HIV patients have enrolled in the treatment. Currently, about 3,702 patients are on ART at the hospital, of whom 3,375 are male. A total of 1,442 HIV-positive adults were nutritionally assessed and determined to be clinically undernourished, with around 980 people receiving therapeutic nutrition [23].

### Source population

All under-nourished HIV positive adults treated with ready-to-use therapeutic food at Debre Markos Referral Hospital.

## Study population

All under-nourished HIV positive adults treated with ready-to-use therapeutic food from the 1st of July, 2015 to the 31st of December, 2017 at Debre Markos Referral Hospital.

## Exclusion criteria

Cases with no charts available at the time of data collection were excluded. Patients with missing baseline data (nutritional status, height, weight, MUAC (for pregnant/bedridden)) were also omitted. Patients who had edema were also omitted since the measurements for the outcome variable would be altered.

## Sample size determination

The sample size was estimated based on the objectives. For estimating the recovery rate a single population proportion formula was used to calculate the sample size by considering the following statistical assumptions:

$$n = \frac{(Z_{\alpha/2})^2 p(1-p)}{d^2}$$

When, n = sample size; d = tolerable marginal error; P = the proportion of nutritional recovery rate from previous study (0.62) [18]. Hence, with 95% confidence level (Z = 1.96) and 5% precision. Finally, after adding 10% for incompleteness of cards, the required sample size for determining the recovery rate was 398. For the second objective, the sample size determination for predictors of treatment recovery rate from nutritional program was calculated using double population proportion formula by using Epi Info version 7 (Centers for Disease Control and prevention) statically programs by considering 95%CI, power 80% and 10% lose to follow up (Table 1). The ration for exposed to non-exposed group is 1 to 1 for this study. Accordingly, the maximum sample size based on the above two formulae after considering 10% incompleteness was 398 and 453. For better estimate of parameters, the maximum sample size of 453 was used.

## Sampling procedure

After computing the minimum adequate sample size, the medical registration numbers of under-nourished HIV positive adults treated with RUTF from the 1st of July, 2015 to the 31st of December, 2017 in the study hospital was recruited. Each patient's record was chosen using a basic random sample procedure with a list of medical registration numbers as sampling frame. The patient charts were chosen using a computer-generated algorithm.

**Table 1. Sample size calculation to assess the recovery rate and its predictors among under-nourished HIV positive adults treated with ready-to-use therapeutic food at Debre Markos Referral Hospital, Northwest Ethiopia, 2018.** (CI:95%, Power = 80%).

| Predictors | Risk ratio | Proportion | | Sample size | Total sample size after adding 10% |
|---|---|---|---|---|---|
| | | Exposed | Un exposed | | |
| Female sex [18] | 1.26 | 67.0 | 53 | 412 | **453** |
| Mild malnutrition [18] | 2.18 | 79.4 | 36.3 | 48 | 52 |
| WHO clinical stage I and II [18] | 1.48 | 83.6 | 56.2 | 100 | 110 |
| Absence of OI [18] | 2.07 | 64.6 | 31.2 | 80 | 88 |

## Study variables

The dependent variable was recovery rate from under nutrition. The independent variables were socio-demographic variables (Age, sex, residency, marital status, religion, employment and educational status); clinical and immunological (CD4 cell count or percentage, Hgb, WHO stages, functional status, ART adherence, types of ART medication regimen anti TB drugs, TB and other opportunistic infections and preventive therapies (CPT and IPT)); anthropometric measures (BMI, MUAC, Weight and Height).

## Definition of study variables

**Event**: Recovered from under-nutrition who was treated with ready–to-use therapeutic food.

**SAM;** HIV positive adults whose baseline BMI $< 16kg/m^2$ or MUAC $<19cm$ for Pregnant/ bedridden [14,24].

**MAM**; HIV positive adults whose baseline BMI relay between $16kg/m^2 \leq BMI < 18.5kg/m^2$ or $19 \leq MUAC < 23$ cm for pregnant/bedridden [14,24].

**Ready- to -use therapeutic food**; is a Plumpynut® -based paste (Plumpy nut or Plumpy sup) in a plastic wrapper for treatment of under nutrition. The ingredients in Plumpy nut or Plumpy sup include "peanut-based paste, with sugar, vegetable, fat and skimmed milk powder, enriched with vitamins and minerals" with energy 2,100 kJ (500) kcal in 92 grams [12].

**Recovered**; under-nourished HIV positive adult reaching BMI of $\geq 18.5$ kg/$m^2$ within the course of therapy or MUAC $\geq$ 23cm for pregnant/bedridden [24].

**Censored;** Patients who were on RUTF but un recovered at the end of the study, dead, and defaulted participants were considered as censored.

**Defaulter**: Participant did not reach a BMI of 18.5 and dropped out of the program before the end of three (MAM) or six (SAM) months [16].

**CD4 count:** was classified as below the threshold (CD4 count $< 200$ cells/$mm^3$ or percentage $<15\%$) and above the threshold (CD4 count $\geq 200$ cells/$mm^3$ or percentage $\geq 15\%$) for severe immunodeficiency [25].

**ART adherence: Good**-if the percentage of missed dose is $> 95\%(< 2$ doses of 30 dose or $<3$ doses of 60 doses); **Fair**-if the percentage of dose between 85–94%(3–5 doses of 30 doses 30 dose or 4–8 doses of 60 dose); **poor**- if dose $< 85\%(> = 6$ doses of 30 doses or $> = 9$ doses of 60 doses) documented by ART physician [25].

## Data collection tool and procedure

This study used secondary data that were collected using structured data extraction checklist. Patient's chart and ART registration book were used as a source of data for this study. At each patient's chart visit, individual HIV and nutritional data were routinely collected on standard-ized forms. The most recent laboratory test results prior to RUTF initiation were used as a baseline value. If there were no laboratory tests registered prior to RUTF intake, results obtained within one month of RUTF initiation were used as a baseline. If two results were obtained within one month, the mean value was used. Data collection tool included socio demographic characteristics, clinical and immunological information, anthropometric mea-surements, RUTF enrolment dates, the presence or absence of OIs diagnosed and NP outcome (categorized as recovered, not recovered, defaulted).

## Data quality control

The data were collected by two experienced ART BSc nurses who were trained on comprehen-sive HIV care and currently involved in the follow up care. The entire data collection process

was closely supervised by one supervisor. One-day training was given for both data collectors and supervisor concerning data extraction checklist and data collection process separately. The filled checklist was checked for their completeness, consistency and accuracy during data management, storage and analysis by the principal investigator.

## Data processing and analysis

The data were entered into Epi-data version 4.2 and analysis was done using STATA version 14. Before analysis, data were cleaned and edited. Results were organized and presented using tables and texts. Patients' cohort characteristics of continuous data were described in terms of central tendency (mean or median), dispersion (standard deviation or inter quartile range) and in frequency distribution for categorical data. The Kaplan-mire survival curve and Log rank test were used to estimate the time to recovery and to compare the survival curves across baseline categorical variables. Both bivariable and multivariable Cox-proportional hazard regression models were fitted to identify independent predictors of recovery rate. In the bivariable analysis, variables having p-value less than or equal to 0.25 were fitted into the multivariable analysis. Finally, in the multivariable analysis variables having p-value less than 0.05 were considered as predictors of recovery. The necessary Cox-proportional hazard regression model assumption was checked by using Schoenfeld residual test. Adjusted hazard Ratio (aHR) with 95% CI were used to report strength of association and statistical significance.

## Ethical considerations

Ethical clearance was obtained from Debre Markos University, Collage of Health Sciences ethical review committee. In addition, formal letter was secured from Debre Markos Referral Hospital administration. As this was a retrospective study, verbal or written consent from the patient was not applicable. To keep confidentiality, the patient names and unique ART number were not included in data collection format. Moreover, the collected data were coded and locked into a separate room before data entry. Furthermore, the data were not disclosed to anyone other than principal investigator. Finally, all the data collected was kept confidential, and would not be used for any other purposes than the stated research objective.

## Results

### Baseline socio-demographic characteristics of patients

The final analysis comprised a total of 453 patient records. More than half of the study participants, 273 (60.3%), were females. The majority of them, 435 (96%) were orthodox Christians, and almost two-thirds,288 (63.6%) came from urban areas. The respondents' average age was 36.20.42 SD years (**Table 2**).

### Clinical, nutritional and HIV/AIDS related characteristics of patients

The RUTF program enrolled a total of 453 HIV-positive persons who were malnourished. At the time of enrolment, 367 (81%) were classed as mild to moderately undernourished. Only around 38 (8.9%) of the study participants were ambulatory and bedridden by functional level, while around 211 (46.6%) had opportunistic infection. The majority of the study subjects,405 (89.4%) exhibited good adherence to ART (**Table 3**).

### Recovery rate of under nourished HIV positive adults treated with RUTF

A total of 453 HIV-positive people on RUTF were observed for varying lengths of time, ranging from 2 to 9 months, with a median follow-up period of 3.94 (IQR:3.64–4.5) months,

**Table 2. Socio-demographic characteristics of participants who were treated with RUTF at Debre Markos Referral Hospital from July of 1st, 2015 –December of 31st, 2017.**

| Variables | Categories | Frequency (N) | Percent (%) |
|---|---|---|---|
| **Gender** | Male | 180 | 39.7 |
| | Female | 273 | 60.3 |
| **Age (years)** | 15–29 | 104 | 23.0 |
| | 30–44 | 257 | 56.7 |
| | ≥45 | 92 | 20.3 |
| **Residence** | Urban | 288 | 63.6 |
| | Rural | 165 | 36.4 |
| **Marital Status** | Single | 68 | 15.0 |
| | Married | 175 | 38.6 |
| | Divorced | 83 | 18.3 |
| | Widowed | 127 | 28.1 |
| **Educational Status** | No formal education | 156 | 34.4 |
| | primary | 105 | 23.2 |
| | Secondary | 125 | 27.6 |
| | Tertiary and above | 67 | 14.8 |
| **Religion** | Orthodox | 435 | 96.0 |
| | Other[†] | 18 | 4.0 |
| **Employment status** | Employed | 81 | 17.9 |
| | Un employed | 372 | 82.1 |

[†] Other religions includes Muslims, Protestants and Catholics.

resulting in a total of 1886.16 person-months observations. Approximately, 201 (44.4%) of the 453 patients participating in the RUTF program were recovered according to the predetermined exit criteria. The overall recovery rate was found to be 10.65 (95% CI: 9.28, 12.23) per 100 person-month observations (**Table 4**).

At four, six, and eight months after starting RUTF, the probability of recovery was 0.65 (95% CI:0.6,0.7), 0.37 (95%CI:0.31,0.43), and 0.25 (95%CI:0.17,0.34), respectively (**Table 5**).

The overall media estimated recovery time was 4.76 months (95% CI: 4.17, 5. 02 months) (**Fig 1**).

The estimated median recovery time was varied depending on the baseline characteristics of cohort patients. Patients with mild to moderate nutritional status at baseline had a faster recovery time, with an estimated recovery time of 4 months (95%Cl: 3.80, 4.98 months), compared to severely undernourished patients, who had an estimated recovery time of 7.75 months (95%CI: 4.89, 7.98 months) (**Fig 2**).

In terms of estimated recovery time by WHO clinical stage, patients with WHO clinical stages I or II had an estimated median recovery time of 4.01 months (95%CI: 3.97, 4.13 months) at the baseline, compared to patients with WHO clinical stages III or IV who had a median recovery time of 7.98 months (95%CI: 6.72, 8.3 months) (**Fig 3**).

## Predictors of recovery time

CD4 count above the threshold, WHO stage I or II, working functional level, mild to moderately undernourished at baseline, absence of OI, and good ART adherence were associated with shorter recovery time in the bivariable Cox-regression. Only three factors were revealed to be predictive of recovery in the multivariable Cox-regression. Patients with WHO clinical

**Table 3. Clinical, nutritional and HIV/AIDS related characteristics patients at Debre Markos Referral Hospital from July of 1st, 2015 to December of 31st, 2017.**

| Variables | | Frequency (N) | Percent (%) |
|---|---|---|---|
| **Baseline nutritional status** | | | |
| | Mild to moderate | 293 | 64.7 |
| | Sever | 160 | 35.3 |
| **WHO clinical stage** | | | |
| | WHO stage I and II | 258 | 57.0 |
| | WHO stage III and IV | 195 | 43.0 |
| **CD4 count** | | | |
| | Below the threshold | 65 | 14.4 |
| | Above the threshold | 388 | 85.6 |
| **Hgb at admission** | | | |
| | <12g/dl | 121 | 26.7 |
| | ≥ 12gm/dl | 332 | 73.3 |
| **Baseline functional status** | | | |
| | Working | 415 | 91.6 |
| | Ambulatory/bedridden | 38 | 8.4 |
| **Experience of OI's** | | | |
| | Yes | 211 | 46.6 |
| | No | 242 | 53.4 |
| **Cotrimoxazole preventive therapy (CPT)** | | | |
| | Yes | 369 | 81.5 |
| | No | 84 | 18.5 |
| **ART adherence** | | | |
| | Good | 405 | 89.4 |
| | Fair/poor | 48 | 10.6 |

**Table 4. Recovery rate at particular time interval of HIV/ADIS adult patients treated with RUTF at Debre Markos Referral Hospital from July of 1st, 2015 to December of 31st, 2017.**

| Follow-up interval | Person-months | Failures | Recovery rate | 95% CI |
|---|---|---|---|---|
| (0–2] | 906.00 | 0 | 0.00 | |
| (2–4] | 780.01 | 136 | 17.43 | 14.73–20.62 |
| (4–6] | 169.25 | 55 | 32.49 | 24.94–42.32 |
| (6–8] | 29.87 | 9 | 30.12 | 15.67–58.89 |
| >8 | 1.02 | 1 | 98.18 | 13.83–69.02 |
| **Total** | **1886.16** | **201** | **10.65** | **9.28–12.23** |

**Table 5. Life table for recovery rate of under nourished HIV positive adults treated with RTUF at 3, 4, 6 and 8 months at Debre Markos referral hospital from July of 1st, 2015 –December of 31st, 2017.**

| Time interval (month) | Beginning total(N) | Recovery status | | Probability of recovery | 95% CI |
|---|---|---|---|---|---|
| | | Recovered(N) | Un recovered(N) | | |
| 3–4 | 453 | 133 | 136 | 0.65 | [0.60,0.70] |
| 4–6 | 184 | 59 | 93 | 0.37 | [0.31,0.43] |
| 6–8 | 32 | 7 | 22 | 0.25 | [0.17,0.34] |
| 8- | 3 | 2 | 1 | 0.05 | [0.001,0.27] |

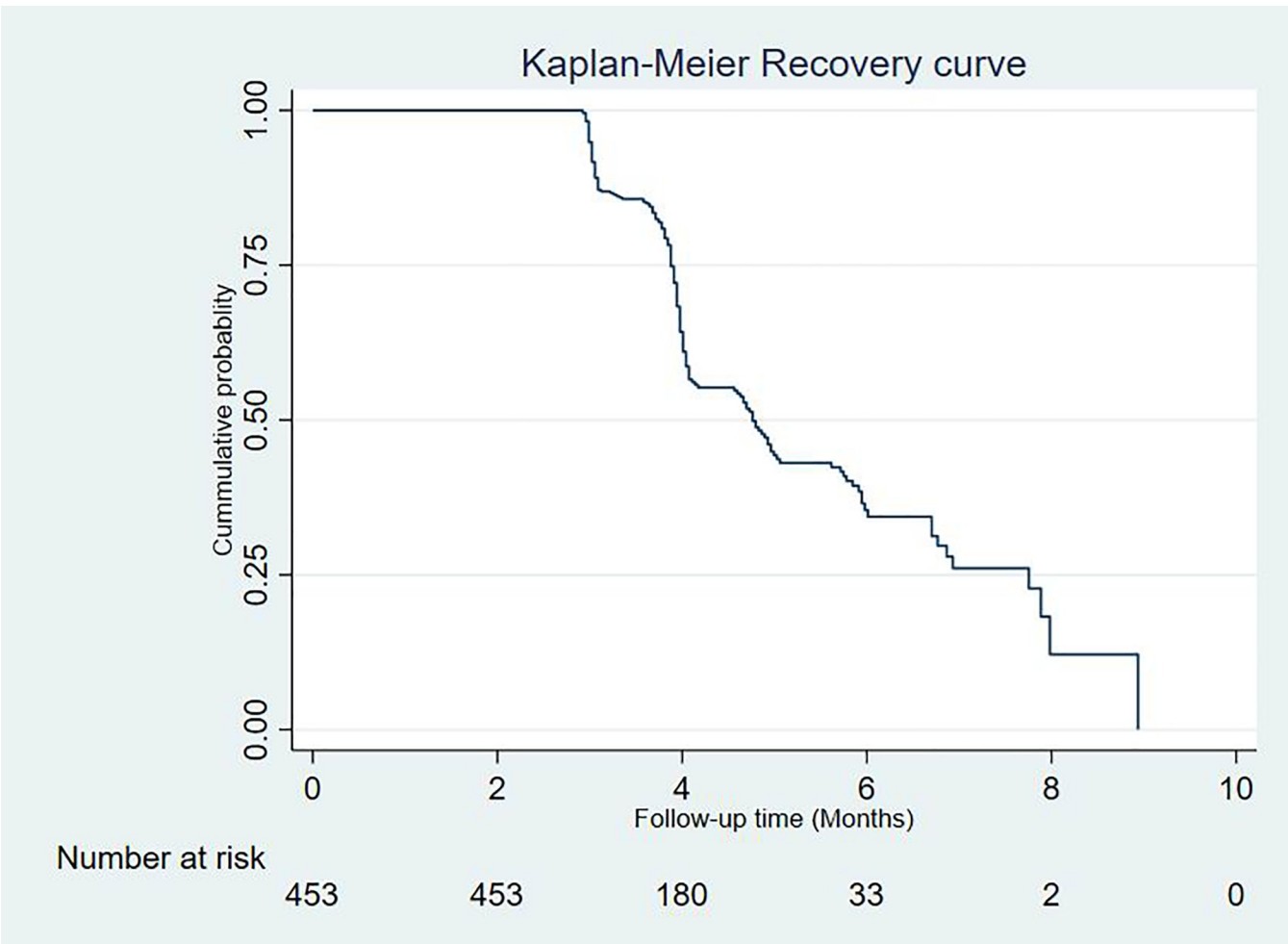

**Fig 1. The overall Kaplan-Meier recovery curve with 95% confidence interval of under nourished HIV positive adults treated with RUTF at Debre Markos Referral Hospital from July 1st, 2015 –December 31st, 2017.**

stage I or II at the time of enrolment had a recovery time that was nearly two times shorter than their counterparts [AHR = 1.99 (95%CI:1.33,2.98)]. Patients with modest to moderate nutritional status at baseline had a nearly four-fold shorter recovery time [AHR = 3.94(95% CI:2.67,5.82)] than severely undernourished patients (**Table 6**).

## Discussion

People living with HIV frequently suffer from malnutrition. Nutritional support in the form of readily to use therapeutic food has been given to these patients on a regular basis to combat the wasting syndrome caused by the chronic debilitating infection [26]. The provision of supplemental foods over considerable duration of time is thought to cause weight gain and improve HIV patients' functioning. As a result, we set out to assess the recovery rate and its determinants among undernourished HIV positive people treated with RUTF at Debre Markos referral hospital in this retrospective follow-up study.

In this study, 44.4%(95%CI: 38.9, 49.0%) of 453 patients participating in the RUTF program were recovered based on predetermined exit criteria. This result was in line with a survey from SSA nations, which found 47.4 percent [19]. However, the recovery rate by percentage

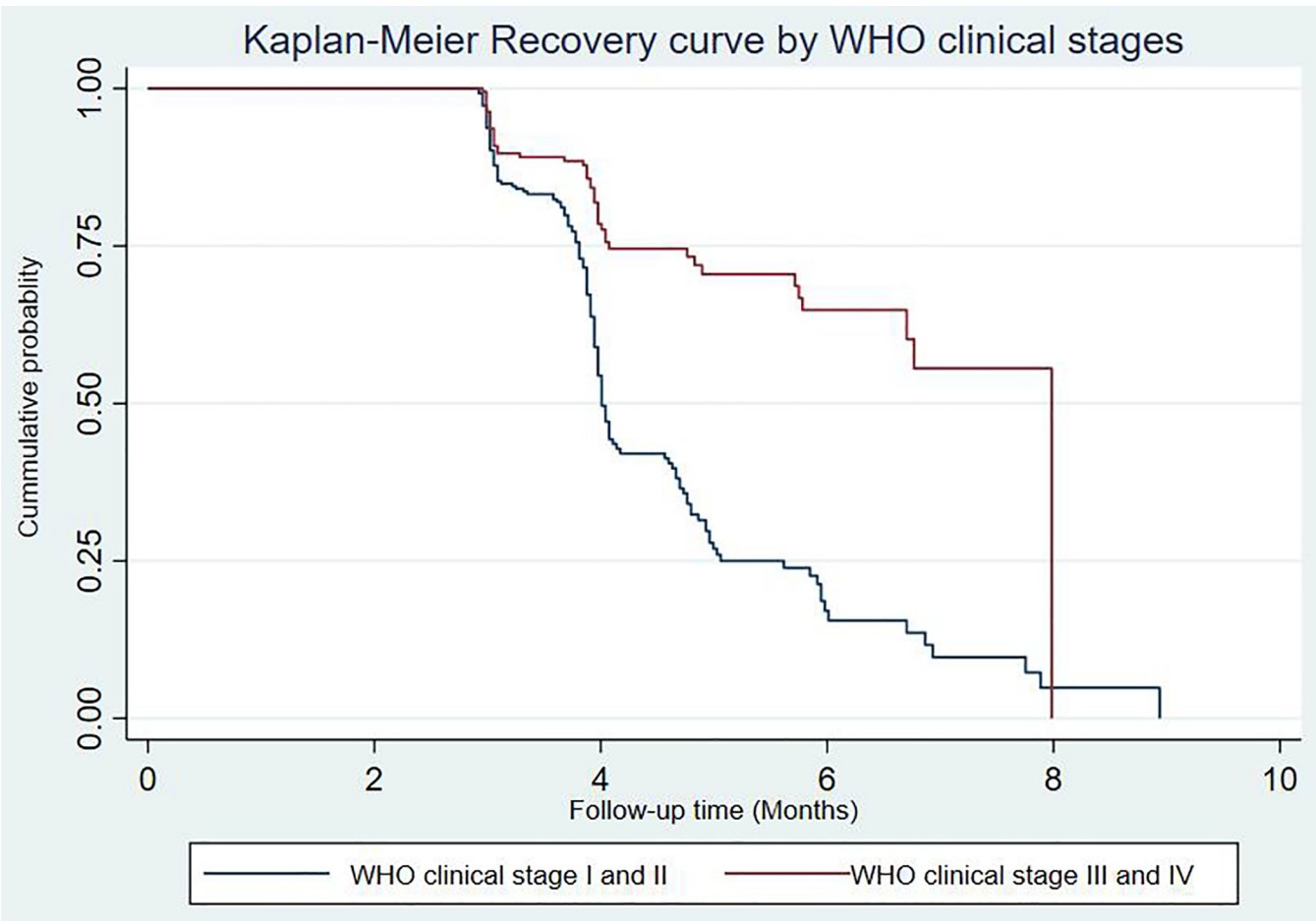

**Fig 2. The estimated Kaplan-Meier recovery curves of under nourished HIV positive adults treated with RUTF by baseline nutritional status at Debre Markos Referral Hospital from July of 1st, 2015 to December of 31st, 2017.**

reported in this study is higher than the 35.3 percent reported in a study from Gondar University Hospital in Northwest Ethiopia [17]. The recovery rate, on the other hand, is lower than the 62.4 percent reported in a research from Mekele Hospital in Northern Ethiopia [18]. Variations in the sample size, the features of the study population, the quality of service offered, nutritional assessment methodologies, research area, and the nutritional and HIV related features of the study participants could all be reasons for the variance in the recovery rate.

The lower recovery rate reported from Gondar University Hospital could be attributable to the fact that the study group included both children and adults, whereas the current study focuses solely on HIV/AIDS-infected adults who are malnourished. Another reason for the higher recovery rate in our study could be because all participants received integrated ART and RUTF services, whereas only 68.1 percent of patients on ART were engaged in RUTF, according to a study conducted in Gondar [17]. This conclusion was supported by results from a randomized control trial research in Africa that looked at the effect of RUTF on mortality in HIV-infected people on ART who were highly immunocompromised [27].

The increased proportion of opportunistic infection in our research participants could be one explanation for the decreased nutritional recovery rate in our study as compared to the study conducted at Mekele. In our study, 46.6 percent of study participants had opportunistic

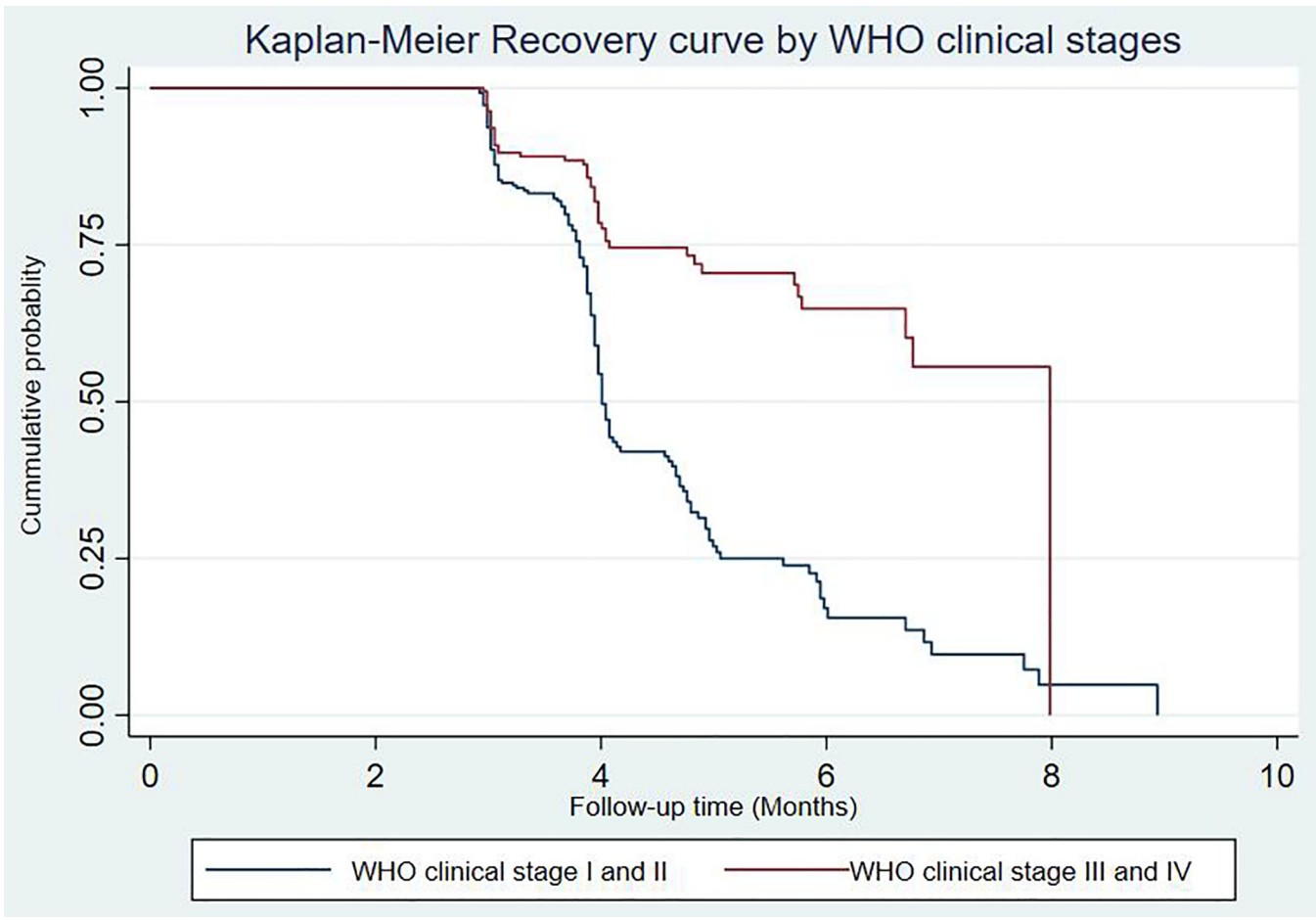

**Fig 3. The estimated Kaplan-Meier recovery of under nourished HIV positive adults treated with RUTF by WHO clinical stages at Debre Markos Referral Hospital from July of 1st, 2015 –December of 31st, 2017.**

infections, whereas only 6.1 percent of study participants had OIs in a research conducted at Mekele Hospital [18].

In terms of nutritional recovery time, the study participants in our study had a median recovery period of 5 months (IQR:3, 8). This finding is consistent with a research out of Kenya that found a median recovery time of 100 days [20]. Another SSA study discovered that the average length of recovery was 3.3 months [19]. The initial reaction to RUTF, on the other hand, was found to be a considerably increased in MUAC value after 4th and 6th months of therapy, according to a study from Gondar University Hospital [17]. The delayed initial response time to recovery at Gondar was due to the method used to assess nutritional recovery to the normal MUAC value. Using MUAC to assess nutritional status for adults may take more time than BMI to reach the normal value. In addition BMI is preferable over MUAC to assess nutritional status for adults though MUAC preferred for special populations like pregnant and bedridden patients [24].

This study identified predictors of recovery time from malnutrition in the study participants. In this study, persons living with HIV who were mild to moderately undernourished at the start had a faster recovery time when treated with RUTF. Previous research conducted somewhere in Ethiopia (a collaboration study by Tufts University and Save the Children) and

**Table 6. Bivariable and multivariable Cox regression of predictors for recovery rate among adults with HIV/AIDS treated with RTUF at Debre Markos Referral Hospital from July 1st, 2015 to December 31st, 2017.**

| Variables | | Nutritional outcomes | | CHR*(95%CI) | AHR**(95%CI) |
|---|---|---|---|---|---|
| | | Recovered (n) | Unrecovered (n) | | |
| Age(years) | 15–29 | 48 | 56 | 1.53(0.97,2.41) | 1.29(0.81,2.05) |
| | 30–44 | 122 | 135 | 1.40(0.94,2.08) | 1.40(0.93, 2.10) |
| | ≥ 45 | 31 | 61 | 1 | 1 |
| Residence | Urban | 136 | 152 | 1.22(0.90,1.64) | 0.86(0.63,1.17) |
| | Rural | 65 | 100 | 1 | 1 |
| Employment | Employed | 46 | 35 | 1.42(1.02,1.98) | 1.18 (0.84, 1.60) |
| | Unemployed | 155 | 217 | 1 | 1 |
| CD4 count | Below the threshold | 23 | 42 | 1 | 1 |
| | Above the threshold | 178 | 210 | 2.03(1.29, 3.19) | 1.40 (0.83, 2.37) |
| WHO clinical stage | stage I or II | 154 | 104 | 3.05(2.19,4.24) | 1.70 (1.13,2.55) |
| | Stage III or IV | 47 | 148 | 1 | 1 |
| Functional status | Working | 194 | 221 | 3.77(1.77,8.04) | 3.33(1.38,8.01) |
| | Ambulatory | 7 | 31 | 1 | 1 |
| Nutritional status | Mild to moderate | 156 | 137 | 4.35 (3.0,6.3) | 3.94 (2.67,5.82) |
| | Sever | 115 | 45 | 1 | 1 |
| Experience of OIs | Yes | 59 | 152 | 1 | 1 |
| | No | 142 | 100 | 2.32(1.71, 3.15) | 1.08 (0.74,1.58) |
| ART adherence | Good | 193 | 212 | 3.39(1.67,6.88) | 1.72 (0.83, 3.54) |
| | Fair/Poor | 8 | 40 | 1 | 1 |

*Crude Hazard Ratio;

**Adjusted Hazard Ratio.

at Mekele hospital backed up this claim [16,18]. Similar findings were also reported from sub-Saharan African countries [19,20]. This shorter recovery time among mild to moderately undernourished patients than severe acutely undernourished patients; could be due to the reason that they possessed a mean BMI of 17.29 kg/m$^2$ (95%CI: 17.23–17.35) which is very close to attain normal nutritional status (BMI≥ 18.5kg/m2) as compared to severe acutely undernourished patients having a mean BMI 15.19kg/m$^2$ (95%CI: 15.02–15.37) which was far away from the normal BMI value and therefore, had lower chance of recovery.

Our research also found that people who were in WHO clinical stage I or II at the baseline had a faster recovery time. A study conducted at Mekele Hospital substantiated this conclusion [18]. End-stage HIV patients typically experience weight loss as a result of the virus. When the WHO clinical stage advances to WHO clinical staging III and IV, the number of comorbidities increases, as does the occurrence of concomitant infections and neoplasms, all of which contribute to a longer recovery period [28]. Furthermore, a study conducted in South Africa revealed that the rate of weight gain in stage IV of HIV infection was much slower than in stage I infection [21].

Furthermore, the current research discovered that having a working functional status at baseline was a substantial predictor of recovery rate. Patients who had a working functional status at the start of the study recovered faster than those who were ambulatory or bedridden. This could be explained by the inability of bedridden and ambulatory patients to take RUTF on time, as well as the incidence of concomitant disease, may cause them to take longer to recover from undernutrition.

This research is not without limitation. The study design, being retrospective cohort may limit our ability to abstract data about predictors that may directly influence the recovery rate from under nutrition, for instance predictors like food sharing at household level, number of persons in the household, additional source of income, daily dose of RUTF prescribed and etc. In addition, the study is a mono center study which may not represent evidence in country level.

## Conclusion

This study concluded that the overall nutritional recovery rate was below the acceptable minimum requirement which at least 75% of patients should recovered. Mild to moderate under-nutrition at baseline, WHO clinical stage I or II at enrolment, and working functional status were found to be predictive of recovery time in HIV/AIDS patients treated with the RUTF. As a result, special attention should be paid to severely malnourished patients, WHO clinical stages III or higher, and patients who are bedridden or ambulatory during treatment. Mild to moderately undernourished at baseline. Furthermore, a prospective cohort study should be done to produce data concerning potential factors that may have a direct impact on recovery, such as food sharing at the household level, family size, supplementary source of income, daily dose of RUTF prescribed, and so on.

## Supporting information

**S1 Dataset. Dataset recovery.**
(DTA)

## Acknowledgments

The authors would like to acknowledge Debre Markos University, and Debre-Markos Referral Hospital for making the data for this research undertaking available. The authors are also grateful to data collectors and supervisors.

## Author Contributions

**Conceptualization:** Habtamu Gebremeskel Woldie.

**Data curation:** Habtamu Gebremeskel Woldie, Animut Alebel.

**Formal analysis:** Habtamu Gebremeskel Woldie, Mulatu Ayana, Animut Alebel.

**Investigation:** Habtamu Gebremeskel Woldie.

**Methodology:** Habtamu Gebremeskel Woldie, Daniel Bekele Ketema, Mulatu Ayana, Animut Alebel.

**Software:** Habtamu Gebremeskel Woldie, Mulatu Ayana, Animut Alebel.

**Supervision:** Habtamu Gebremeskel Woldie, Daniel Bekele Ketema, Mulatu Ayana, Animut Alebel.

**Validation:** Habtamu Gebremeskel Woldie, Mulatu Ayana, Animut Alebel.

**Visualization:** Habtamu Gebremeskel Woldie, Daniel Bekele Ketema, Mulatu Ayana, Animut Alebel.

**Writing – original draft:** Habtamu Gebremeskel Woldie, Daniel Bekele Ketema, Mulatu Ayana, Animut Alebel.

**Writing – review & editing:** Habtamu Gebremeskel Woldie, Daniel Bekele Ketema, Mulatu Ayana, Animut Alebel.

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
