## [Decision Letter · Decision Letter 0]

30 May 2021

PONE-D-21-05739

Predictors of recovery among undernourished HIV-positive adults treated with ready-to-use therapeutic food at Debre Markos Comprehensive Specialized Hospital: a retrospective cohort study

PLOS ONE

Dear Dr. Daniel Bekele Ketema,

Thank you for submitting your manuscript to PLOS ONE. After careful consideration, we feel that it has merit but does not fully meet PLOS ONE’s publication criteria as it currently stands. Therefore, we invite you to submit a revised version of the manuscript that addresses the points raised during the review process.

Kindly ensure that you make all the data underlying the findings in the manuscript fully available. Note that the PLOS Data policy requires authors to make all data underlying the findings described in their manuscript fully available without restriction, with rare exception (please refer to the Data Availability Statement in the manuscript PDF file). The data should be provided as part of the manuscript or its supporting information, or deposited to a public repository. For example, in addition to summary statistics, the data points behind means, medians and variance measures should be available. If there are restrictions on publicly sharing data—e.g. participant privacy or use of data from a third party—those must be specified.

Also, ensure that the manuscript is written in standard English. PLOS ONE does not copyedit manuscripts, so the language in submitted articles must be clear, correct, and unambiguous. Any typographical or grammatical errors should be corrected at revision.

We look forward to receiving your revised manuscript.

Kind regards,

Obinna Ikechukwu Ekwunife, PhD

Academic Editor

PLOS ONE

Journal Requirements:

2. In ethics statement in the manuscript and in the online submission form, please provide additional information about the patient records used in your retrospective study. Specifically, please ensure that you have discussed whether all data were fully anonymized before you accessed them and/or whether the IRB or ethics committee waived the requirement for informed consent. If patients provided informed written consent to have data from their medical records used in research, please include this information.'

Reviewers' comments:

Reviewer's Responses to Questions

**Comments to the Author**

1. Is the manuscript technically sound, and do the data support the conclusions?

Reviewer #1: Yes

Reviewer #2: Yes

2. Has the statistical analysis been performed appropriately and rigorously? 

Reviewer #1: Yes

Reviewer #2: Yes

3. Have the authors made all data underlying the findings in their manuscript fully available?

Reviewer #1: No

Reviewer #2: No

4. Is the manuscript presented in an intelligible fashion and written in standard English?

Reviewer #1: No

Reviewer #2: Yes

5. Review Comments to the Author

Reviewer #1: Manuscript PONE-D-21-05739. The article on "Predictors of recovery among malnourished adults with HIV treated with ready-to-use therapeutic foods" is interesting as it addresses nutritional support, which is increasingly recognized as an important part of the care package for people living with HIV/AIDS. The aim of this research is therefore to estimate the recovery rate and predictors among malnourished adults with HIV/AIDS treated with therapeutic foods.

To improve its work, it is suggested to.

1. In the Abstract, include a paragraph of the contribution of this research and future work.

2. Review the English grammar and spelling of the entire paper.

3. Indicate whether this research uses publicly available information or has a data set with access to the data for researchers to replicate the experiment.

4. To make your research replicable I suggest placing the data from the study with public access; you can create one at https://data.mendeley.com/

5. Improve the quality and resolution of the figures.

6. In the Conclusions section include a paragraph about future work to be done.

Reviewer #2: Abstract

Line 41, “clinical stage” should be specific. So it should be modified as “ WHO clinical stage”

Line 44-45, this study found that the overall nutritional recovery rate was below the World Health Organization recommended standard. Please state the recommended standard rate.

Introduction

Why you mentioned the specific category of some variables (WHO stage I and II, absence of opportunistic infection, CD4 cells percentage above the threshold)?

The introduction should show the gap of previous studies

Methods

Line 114-117, “Currently, a total of 9,612 people living with HIV had ART follow up at Debre Markos referral hospital. Of whom, about 3,702 people had active ART follow up among these 3,375 were adults”. It is not clear. What is the difference between “Current ART follow up=9,612” and “active ART follow up=3702”?

The eligibility criteria should be stated separately and clearly

Did you include patients with edema (might affect the measurement accuracy of outcome variable)

During sample size calculation, what was the ratio of unexposed to exposed group in each variables? You should state clearly

Line 170-171, I hope the follow up period for MAM and SAM patients are completely different. If so, how can diagnose default or censored with the same follow up period?

Who were classified as “defaulted”? Are they similar to participant who didn’t experience the event at the end of the study?

Line 180-181, the data source is not clear. Did you collect the data only from patients chart? Is there any other sources? To avoid information bias, it is advised to use different data sources.

Line 208: Adjusted hazard Ratio (aHR) wand 95% CI………..” change the word “ wand” to “ with”

Result and discussion

Line 280-282, participants were classified as mild, moderate and severe based on their BMI which was also your outcome variable. So, a patient with high BMI (near to discharge criteria) at baseline is expected to gain the target BMI faster than a patient with extremely low BMI. No need of research. It was better to do stratified analysis (mild, moderate or severe)

In table 6, why did you take the risk groups (rural, WHO clinical stage III or IV, Unemployed….) as a reference category? I hope your research problem is long recovery time and all the above categories are risk for your problems.

Line 360, “number of household” is not clear.

6. PLOS authors have the option to publish the peer review history of their article (what does this mean?). If published, this will include your full peer review and any attached files.

Reviewer #1: No

Reviewer #2: No

---

## [Author Response · Author response to Decision Letter 0]

10 Jun 2021

First: We really appreciate the editor and reviewers, and we would like to express our thanks for devoting their time and look at each and every word of the article.

Review Comments to the Author

Reviewer #1: Manuscript PONE-D-21-05739. The article on "Predictors of recovery among malnourished adults with HIV treated with ready-to-use therapeutic foods" is interesting as it addresses nutritional support, which is increasingly recognized as an important part of the care package for people living with HIV/AIDS. The aim of this research is therefore to estimate the recovery rate and predictors among malnourished adults with HIV/AIDS treated with therapeutic foods.

To improve its work, it is suggested to.

1. In the Abstract, include a paragraph of the contribution of this research and future work. Response: Thank you for suggestion. As per your suggestion we have included the significance of this study for concerned body and future researcher (Check Abstract section line # 26-30, page 2)

2. Review the English grammar and spelling of the entire paper.

Response: Thanks for your comments: We extensively reviewed the manuscript for grammar and spelling error (Check the clean manuscript)

3. Indicate whether this research uses publicly available information or has a data set with access to the data for researchers to replicate the experiment.

Response: We provide a Data Availability Statement (“All relevant data are within the manuscript and its Supporting Information files”) on the PLOS ONE submission system

4. To make your research replicable I suggest placing the data from the study with public access; you can create one at https://data.mendeley.com/

Response: thank you for your recommendation: We prefer to set all relevant data within the manuscript and its Supporting Information. Therefore, we submit the data set as a supporting file with a data set name: Dataset_Recovery. 

5. Improve the quality and resolution of the figures.

Response: As per your direction we improved the quality and resolution of the figures 

(Check Fig 1, Fig 2, and Fig 3)

6. In the Conclusions section include a paragraph about future work to be done

Response: Thank you; We include what should be done in the future (Check Conclusion section line# 369-371 page21)

Reviewer #2: Abstract

Line 41, “clinical stage” should be specific. So it should be modified as “WHO clinical stage”

 Response: thank you for your suggestion: We have corrected it as per your suggestion 

(Check Abstract section line#49, page 3)

Line 44-45, this study found that the overall nutritional recovery rate was below the World Health Organization recommended standard. Please state the recommended standard rate.

Introduction

Response: We have stated the recommended WHO standard (Check Abstract section, Line#48)

Why you mentioned the specific category of some variables (WHO stage I and II, absence of opportunistic infection, CD4 cells percentage above the threshold)?

 Response: we mentioned specific categories of variables. This is recommending to highlight advantages and dis advantageous groups in terms of outcome of interest. This intern helps for researcher, policy makers and programmer to set a priority and plan for intervention. 

The introduction should show the gap of previous studies

Response: We have indicated identified gaps of existing literatures and highlighted how this research fill that gaps (check Introduction section, Line# 88-91, Page 5) 

Methods

Line 114-117, “Currently, a total of 9,612 people living with HIV had ART follow up at Debre Markos referral hospital. Of whom, about 3,702 people had active ART follow up among these 3,375 were adults”. It is not clear. What is the difference between “Current ART follow up=9,612” and “active ART follow up=3702”?

Response: Thank you for insightful comments. The idea of the statement is to highlight background information about total number of patients enrolled on ART since started and till now. Therefore, we revised this statement to avoid such confusions for the readers (Check Method section, Line #113-117, page 6). 

The eligibility criteria should be stated separately and clearly

Response: Check (Method section, Line # 124-128, page 6-7)

Did you include patients with edema (might affect the measurement accuracy of outcome variable)?

Response: Thank you for your comments, we have already excluded patients presented with edema (Check Method section, Line #127-128, Page 7)

During sample size calculation, what was the ratio of unexposed to exposed group in each variable? You should state clearly

Response: thank you for this comment. Ratio of unexposed to exposed depends upon the outcome of interest and study population. In our case the ration of exposed to un exposed for the study population is 1 to 1 (Check Method section, Line # 140-141, page 7)

Line 170-171, I hope the follow up period for MAM and SAM patients are completely different. If so, how can diagnose default or censored with the same follow up period?

Response: Regardless of their bassline nutritional status either MAM or SAM, participants who were not fully contributed for outcome variable were considered as censored. The follow up period could not have expected to be similar for MAM and SAM patients. For this study patients were considered us censored when the end point of the interest (Recovery) has not been observed due to certain reason like

Patients still on treatment (on RUTF) but un recovered at the end of the study

The recovery status of patients at the time of analysis might not be known because of 

o Default (lost follow-up)

o Death 

Who were classified as “defaulted”? Are they similar to participant who didn’t experience the event at the end of the study?

Response: In this study patients were classified as defaulted when they did not reach a BMI of 18.5 and dropped out of the program before the end of three (MAM) or six (SAM) months follow up (Check Method section, Line # 174-175, page 9). There is an important assumption made in survival analysis to make appropriate use of the censored data. This is called non-informative censoring and essentially assumes that the participants whose data are censored (defaulted) would have the same distribution of recovery time if they were actually observed. Therefore, those who are defaulters have similar event distribution with patients whose outcome is observed. 

Line 180-181, the data source is not clear. Did you collect the data only from patient’s chart? Are there any other sources? To avoid information bias, it is advised to use different data sources.

Response: Thank you for your comment: In addition to patients chart we have used patient’s registration book as source of data for this study (Check method section, line # 184-185, page 10)

Line 208: Adjusted hazard Ratio (aHR) wand 95% CI………..” change the word “ wand” to “ with”

Response: we have addressed this typographical error (Check method section, line # 213, page 11)

Result: 

Result and discussion

Line 280-282, participants were classified as mild, moderate and severe based on their BMI which was also your outcome variable. So, a patient with high BMI (near to discharge criteria) at baseline is expected to gain the target BMI faster than a patient with extremely low BMI. No need of research. It was better to do stratified analysis (mild, moderate or severe)

Response: Thank you for this comments which allow us to elaborate more

The baseline nutritional status (mild, moderate and severe) is not our outcome variable rather used for adjusting RUTF dose and follow up time. Our outcome variable is time to recovery after initiating this therapeutic feeding. The conclusion you made “a patient with high BMI (near to discharge criteria) at baseline is expected to gain the target BMI faster than a patient with extremely low BMI” might be true. This information is supported by many existing literatures. But it did not mean no need of conduct research in this area. In the scientific world, we can support the existing evidence with available data on hand to facilitated evidence based decision by providing information from different segments. That means exploring this fact at particular study site is important. In addition, quantifying this fact after adjusting with different potential confounders which were not addressed before may produce parsimonious estimate for population parameter. Moreover, this known fact by itself will be used for adjusting other potential predictor variables. 

In table 6, why did you take the risk groups (rural, WHO clinical stage III or IV, Unemployed….) as a reference category? I hope your research problem is long recovery time and all the above categories are risk for your problems.

Response: thank you for this important comments. It’s best to choose a category as a reference that makes interpretation of results easier. As you stated above long recovery time is a problem for our study; and selecting non exposed group as a reference give a sense with easy interpretation. However, it is possible to interpreting our results in terms of accelerated time. That means those patients who were in the non-exposed group have accelerated recovery time compared to those who are exposed. Moreover, we choose these categories as a reference is for comparison purpose with other similar studies. 

Note: If our justifications were not convincing we are ready to accept your suggestion 

Line 360, “number of household” is not clear.

Thank you: we have corrected it as “Number of persons in the household” (Check discussion section, line # 359-360, page 21)

---

## [Decision Letter · Decision Letter 1]

19 Jul 2021

Predictors of recovery rate among undernourished HIV-positive adults treated with ready-to-use therapeutic food at Debre Markos Comprehensive Specialized Hospital: a retrospective cohort study

PONE-D-21-05739R1

Dear Dr. Daniel Bekele Ketema,

We’re pleased to inform you that your manuscript has been judged scientifically suitable for publication and will be formally accepted for publication once it meets all outstanding technical requirements.

Kind regards,

Obinna Ikechukwu Ekwunife, PhD

Academic Editor

PLOS ONE

Additional Editor Comments (optional):

Reviewers' comments:

Reviewer's Responses to Questions

**Comments to the Author**

1. If the authors have adequately addressed your comments raised in a previous round of review and you feel that this manuscript is now acceptable for publication, you may indicate that here to bypass the “Comments to the Author” section, enter your conflict of interest statement in the “Confidential to Editor” section, and submit your "Accept" recommendation.

Reviewer #1: All comments have been addressed

2. Is the manuscript technically sound, and do the data support the conclusions?

Reviewer #1: Yes

3. Has the statistical analysis been performed appropriately and rigorously? 

Reviewer #1: Yes

4. Have the authors made all data underlying the findings in their manuscript fully available?

Reviewer #1: Yes

5. Is the manuscript presented in an intelligible fashion and written in standard English?

Reviewer #1: Yes

6. Review Comments to the Author

Reviewer #1: The article has been significantly improved, the reviewers' suggestions have been implemented. I suggest it be considered for publication.

7. PLOS authors have the option to publish the peer review history of their article (what does this mean?). If published, this will include your full peer review and any attached files.

Reviewer #1: No

---

## [Editor Report · Acceptance letter]

23 Jul 2021

PONE-D-21-05739R1 

Predictors of recovery rate among undernourished HIV-positive adults treated with ready-to-use therapeutic food at Debre Markos Comprehensive Specialized Hospital: a retrospective cohort study 

Dear Dr. Ketema:

I'm pleased to inform you that your manuscript has been deemed suitable for publication in PLOS ONE. Congratulations! Your manuscript is now with our production department. 

Kind regards, 

on behalf of

Dr. Obinna Ikechukwu Ekwunife 

Academic Editor

PLOS ONE